# Selenium Nanomaterials Enhance the Nutrients and Functional Components of Fuding Dabai Tea

**DOI:** 10.3390/nano14080681

**Published:** 2024-04-15

**Authors:** Xiaoli Zhang, Xiaona Li, Feiran Chen, Xuesong Cao, Chuanxi Wang, Liya Jiao, Le Yue, Zhenyu Wang

**Affiliations:** 1Institute of Environmental Processes and Pollution Control, School of Environment and Ecology, Jiangnan University, Wuxi 214122, China; 2Jiangsu Engineering Laboratory for Biomass Energy and Carbon Reduction Technology, Wuxi 214122, China

**Keywords:** selenium nanomaterials, theanine, tea polyphenols, regulatory mechanism

## Abstract

Theanine, polyphenols, and caffeine not only affect the flavor of tea, but also play an important role in human health benefits. However, the specific regulatory mechanism of Se NMs on fat-reducing components is still unclear. In this study, the synthesis of fat-reducing components in Fuding Dabai (FDDB) tea was investigated. The results indicated that the 100-bud weight, theanine, EGCG, total catechin, and caffeine contents of tea buds were optimally promoted by 10 mg·L^−1^ Se NMs in the range of 24.3%, 36.2%, 53.9%, 67.1%, and 30.9%, respectively. Mechanically, Se NMs promoted photosynthesis in tea plants, increased the soluble sugar content in tea leaves (30.3%), and provided energy for the metabolic processes, including the TCA cycle, pyruvate metabolism, amino acid metabolism, and the glutamine/glutamic acid cycle, ultimately increasing the content of amino acids and antioxidant substances (catechins) in tea buds; the relative expressions of key genes for catechin synthesis, *CsPAL*, *CsC4H*, *CsCHI*, *CsDFR*, *CsANS*, *CsANR*, *CsLAR*, and *UGGT*, were significantly upregulated by 45.1–619.1%. The expressions of theanine synthesis genes *CsTs*, *CsGs*, and *CsGOGAT* were upregulated by 138.8–693.7%. Moreover, Se NMs promoted more sucrose transfer to the roots, with the upregulations of *CsSUT1*, *CsSUT2*, *CsSUT3*, and *CsSWEET1a* by 125.8–560.5%. Correspondingly, Se NMs enriched the beneficial rhizosphere microbiota (*Roseiarcus*, *Acidothermus*, *Acidibacter*, *Conexicter*, and *Pedosphaeraceae*), enhancing the absorption and utilization of ammonium nitrogen by tea plants, contributing to the accumulation of theanine. This study provides compelling evidence supporting the application of Se NMs in promoting the lipid-reducing components of tea by enhancing its nitrogen metabolism.

## 1. Introduction

Tea, a popular non-alcoholic beverage with unique flavor, is prepared by the picking, spreading, fixing, rolling, and drying of fresh shoots from the tea (*Camellia sinensis* (L.)) bushes. According to the statistics of the Food and Agriculture Organization of the United Nations (FAO, 2018), global production of tea increased by 4.4% annually from 2008 to 2018 and is expected to exceed 6 million tonnes per year by 2027 (https://www.fao.org/statistics/en/, accessed on 29 May 2018). The bioactive compounds such as theanine [1], tea polyphenols [2], and caffeine taken obtained through the process of long-term tea-drinking can reduce cardiovascular and cerebrovascular disease, reduce cholesterol and blood pressure, and alleviate anxiety [3,4]. The World Health Organization (WHO, 1997) recognized obesity as a global chronic epidemic and a major public health problem to be addressed [5]. Lin et al. found that by feeding animals black tea, oolong tea, pu-erh, and green tea leaves, the weight of rats, along with their plasma triglycerides, cholesterol, and low-density lipoprotein cholesterol, were significantly decreased [6]. This is mainly because the effective components abundantly present in tea, such as theanine, caffeine, tea polyphenols, and dietary fiber, could prevent the expression of fat synthesis-related enzymes, promote the oxidation of fatty acids, suppress appetite, and inhibit the absorption of nutrients [7]. Therefore, increasing the content of metabolites such as theanine, tea polyphenols, caffeine, and other key quality factors in green tea provides great potential for the consumption of tea to become a healthy and effective way to reduce fat. In the past, tea quality and yield were mainly improved through the massive application of chemical fertilizers [8]. These excessive nutrient inputs would lead to serious environmental problems, such as soil acidification, nutrient loss, harmful metal deposition, and agricultural surface pollution [9,10]. Therefore, a green, safe, and efficient method for increasing tea production and improving tea quality is urgently needed to enhance the competitiveness and economic benefits of the tea market.

Nanomaterials (NMs) are microscopic materials with particle sizes between 1 and 100 nm. The physical and chemical properties of NMs, including crystallinity, catalytic activity, porosity, and aggregation, give materials the ability to cross the cell barrier to reach different tissues and organs of the plant [11]. The internalized NMs were proven to improve photosynthetic rate, enhance antioxidant capacity and secondary metabolism, and regulate related genes to improve crop yield and quality [12,13,14]. The application of NMs can also reduce toxicity and minimize possible negative impacts on the environment (such as groundwater pollution) [15]. NMs, with their trace elements, often show better physical and chemical properties than ordinary materials. Selenium (Se) is a trace element necessary for human metabolism, with a variety of health benefits such as antioxidant, antibacterial, cytoprotective, anti-inflammatory, and anti-cancer activities [16,17]. In addition, low concentrations of selenium have beneficial effects on plant growth, yield, and nutritional quality improvement. At present, research regarding the role of Se NMs in facilitating plant growth and quality, as well as improving human metabolism, has been reported. For example, Li et al. found that spraying 10 mg·L^−1^ Se NMs on the tea plants can significantly increase the content of soluble sugar, protein, carotenoids, tea polyphenols, and catechins in tea, as well as promote its secondary metabolism [18]. The foliar application of Se NMs elevated the contents of chlorophyll and soluble sugar and activated the pathways of phenylpropane and branched chain fatty acid, as well as the expression of related enzymes and genes, in pepper fruits, resulting in the enhanced synthesis of flavonoids, total phenols, and capsaicin [19,20]. Theanine, as a unique free amino acid in tea, contributes to its special umami or flavor and counters the astringency and bitterness of tea infusions [21,22], contributing to protection against certain cancers and cardiovascular diseases and promoting weight loss and immune system performance [1]. Tea is also rich in catechins, which are beneficial to human health.

Therefore, we hypothesized that the foliar application of Se NMs could significantly improve the theanine, tea polyphenols, caffeine, and other quality contents, thereby improving the fat reduction properties of green tea. Therefore, this study aims to explore the optimal concentration of Se NMs needed to improve the nutritional level of green tea, as well as the biological responses and mechanisms of nutrients and functional components mediated by Se NMs in tea. The results provide key information for the use of Se NMs to improve the quality of green tea, thereby establishing efficient and sustainable nano-agricultural technology.

## 2. Materials and Methods

### 2.1. Synthesis and Characteristic of Se NMs

All the chemical reagents employed were analytic grade reagents (AR) purchased from Sinopsin Group Chemical Reagent Co., Ltd. (Shanghai, China). The synthesis process of Se NMs was conducted using the method previously published in Refs. [23,24]. For the preparation of Se NMs, a mixed solution of 4 × 10^−5^ M selenic acid (H_2_SeO_3_, ≥95%) and pre-made raisin extracts were heated under reflux conditions (pH 5.9). The mixture was centrifuged at 17,280× *g* for 20 min, and the sediments were used to obtain Se NMs. The shape and size distribution of Se NMs were characterized by TEM (JEM-2100, Japan Electronics Corporation, Tokyo, Japan), and the TEM images were used to determine the average particle size through direct measurement of 50 particles. A Zetasizer (Nano-ZS90, Malvern Instruments Ltd., Malvern, UK) device was used to obtain the hydrodynamic diameter and for zeta potential analysis (Appendix A). The allotropic composition was identified using X-ray diffraction (XRD, D8 Advance, Bruker, Karlsruhe, Germany). The Se valence of Se NMs was tested using X-ray photoelectron spectroscopy (XPS, ESCALAB 250XI, Thermo Fisher Scientific, Waltham, MA, USA).

### 2.2. Field Experimental Design and NMs Exposure

The experimental field was situated at Wuxi, Jiangsu (31.52° N, 120.14° E), and tea trees (*Camellia sentences* L. cv. Fuding Dabai) were used as the experimental material; each experimental field plot covered 2 m^2^. Before the spring bud sprouting, the field experiments were conducted using different Se NMs treatments, with foliar suspensions applied every 2 days (four applications in total) in March of 2021. The selected time was about 16:00, under fair weather conditions, ensuring that there was no chance of rain for 4 h after spraying. The spraying concentration of Se NMs comprised four treatments, including 0 (CK, the blank control group, i.e., the tea plants without the application of Se NMs), 5, 10, and 50 mg·L^−1^, respectively. Moreover, due to the fact that the yield and quality of the FDDB tea bud is highest with the application of 10 mg·L^−1^ Se NMs, an ion control (equivalent Se with 10 mg·L^−1^ Se NMs) and a common conventional commercial fertilizer containing 46% Na_2_SeO_3_ was included. According to the experimental results (Appendix A), the final contents of theanine, catechins, and caffeine in the tea buds were significantly increased after Se NMs exposure compared to those of the unexposed control group, and the theanine and caffeine contents were not significantly different between the unexposed group and the ion group. During the experimental period regarding the normal growth of tea plants, there were no additional management measures employed to reduce other interfering factors. After the spraying period of 14 days, pooled leaf samples with one bud and one leaf were picked from each plot. Meanwhile, the soils were collected from each experimental tea garden on 26 March 2021.

Before sampling, the photosynthesis parameters of Fuding Dabai leaves were obtained using a CIRAS-3 (PP-Systems Company, Amesbury, MA, USA) portable gas-exchange system. The contents of photosynthetic pigment were measured according to the method published in Ref. [25]. The theanine, catechins (HPLC ≥ 98%), soluble sugar, glutamine synthetase (GS), glutamate synthetase (GOGAT), ammonium nitrogen, and nitrate nitrogen standards assay kits were obtained from Solarbio (Beijing, China), and the caffeine standard (purity in 98–99%) was purchased from Zhongke Quality Inspection Bio Co., Ltd. (Beijing, China).

### 2.3. Mineral Elements Analysis after Se NMs in Tea Buds

After different treatments, the contents of mineral nutrients in tea buds were determined by inductively coupled plasma mass spectrometry (ICP-MS, iCAP-TQ, Thermo Fisher, Dreieich, Germany). In brief, 25 mg dried samples were digested by adding 3 mL HNO_3_ (65–68%) and 3 mL ultrapure water for microwave digestion at 190 °C and 1400 W for 30 min. For the accuracy tests, a standard reference (bush branches and leaves, GBW07602) was simultaneously digested and analyzed by ICP-MS.

### 2.4. Theanine, Catechins, and Caffeine Analysis and Metabolic Response of Tea Buds

The components of theanine, catechins, and alkaloids were detected by high-performance liquid chromatography–tandem mass spectrometry (HPLC–MS/MS, Thermo Scientific, Dreieich, Germany). Briefly, the fresh tea buds were ground into a powder, 0.1 g of ground tea buds was added to 50 mL of DI water and then heated to 100 °C for 30 min in a water bath. After being cooled to room temperature, the solution was passed through a 0.22 μm water film into an autosampler vial for theanine analysis. To 0.1 g of the powder samples, 10 mL of 70% methanol solution was added, and the mixture was then heated to 70 °C in a water bath for 15 min. The supernatants were centrifuged at 6000 rpm for 5 min, re-extracted, and mixed. The obtained supernatants were filtered using an organic membrane (pore size: 0.22 μm) into an autosampler vial to detect the contents of catechins and caffeine. The HPLC–MS/MS parameters are set up according to the methods of Li et al. and are described in Text S1 [18].

The metabolites in the tea buds were also determined by HPLC–MS/MS. The fresh tea tissue samples were immediately frozen, and 100 mg of the samples were homogenized with liquid nitrogen, placed into a 2 mL centrifuge tube, and then extracted using 1.5 mL methanol–water (4:1) solution. This mixture was set in an ice water bath to finish the ultrasonic extraction (30 min, 35 kHz). After that, the mixture was centrifuged at 12,000 rpm at 4 °C for 15 min. The supernatant was further dried in a vacuum with a rotary evaporation concentrator (4 °C) and redissolved with 200 μL methanol acetonitrile–water, before again being centrifuged at 12,000 rpm at 4 °C for 10 min. The parameters of HPLC–MS/MS are added in Text S1 [26,27].

### 2.5. Tea Soil Microbial Community Structure Analysis Based on 16S rRNA Gene Sequencing

To characterize the diversity and composition of the soil bacterial community, high-throughput sequencing of the bacterial 16S rRNA gene was conducted using the Illumina MiSeq platform. The total DNA of the microorganisms was extracted from the soil samples, and then the V3–V4 region of amplification was achieved using the quantitative PCR system (ABI, Carlsbad, CA, USA). The sequencing library was constructed with the Illumina’s TruSeq Nano DNA LT Library Prep Kit (San Diego, CA, USA) and using 2% agarose gel electrophoresis. Before the high-throughput sequencing, the library quality was confirmed using the Agilent High Sensitivity DNA Kit and quantified using the Quant-it PicoGreen dsDNA Assay Kit. Finally, 250 bp paired-end reads were sequenced on the Illumina MiSeq platform with MiSeq reagent kit v3 from Shanghai Personal Biotechnology Co., Ltd. (Shanghai, China).

### 2.6. Related Gene Expression of Quantitative Real-Time (qPCR) Analysis

The expression of genes (*GS*, *GOGAT*, *CsTS*, *CsPAL*, *CsC4H*, *CsCHI*, *CsANR*, *CsLAR*, *CsANS*, *CsDFR*, *UGGT*, *CsTCS*, *SUT*, and *SWEET*) associated with theanine, catechins, caffeine synthesis, and carbohydrate transport in tea buds foliar-treated with 10 mg·L^−1^ Se NMs were investigated by qPCR. The total RNA was extracted from the tea buds using a MiniBEST Plant RNA Extraction Kit (Takara, Japan) and was reverse transcribed into first-strand cDNA. Each qPCR amplification was carried out with 50-µL reaction volumes consisting of 1 µL forward primer, 1 µL reverse primer, 25 µL UltraSYBR mixture, 2 µL cDNA template, and 21 µL distilled deionized water. The PCR cycling conditions were as follows: initial denaturation was performed for 30 s at 95 °C, followed by 40 cycles of denaturation for 5 s at 95 °C and extension for 30 s at 60 °C. At the end of the PCR reaction, a melting curve was generated. For each biological sample, the program was performed in triplicate using a CFX96™ real-time system (BioRad, Hercules, CA, USA). Relative gene expressions were calculated using the 2^−ΔΔCt^ method. A list of the specific primers used in this study is presented in Appendix A.

### 2.7. Statistical Analysis

The measured data from all chemical analyses were presented as means ± standard deviation (at least three independent replicates). One-way ANOVA and LSD tests were applied to demonstrate the significance of difference using SPSS (IBM Corporation, version 26.0, Armonk, NY, USA), and *p* < 0.05 was considered statistically significant. The analyses of sequence data were performed using QIIME2 and R packages (v3.5.0). Column charts were plotted with Origin 2021.

## 3. Results and Discussion

### 3.1. Characterization of Se NMs

The size of the Se NMs ranged from 20 to 90 nm (Figure 1A), with an average size of 62.3 ± 9.9 nm (Figure 1B). The hydrodynamic diameter of the Se NMs in DI water was 156.8 ± 9.9 nm, while the zeta potential was −24.0 ± 1.6 mV (Appendix A). A very broad feature of the XRD pattern (Figure 1C) indicates that the product of the peak shape accords with the standard Se NMs. The results of the XPS assay indicated that the Se 3d spectra can be deconvoluted into two peaks at 54.9 eV and 55.7 eV (uncorrected), corresponding to the Se 3d3/2 and Se 3d5/2 components in Se NMs, respectively (Figure 1D). The results were consistent with those of the previous study [27]. The plant stomata size is between 10–100 μm, and NMs can be absorbed when the leaves are fully open [28]. Previous studies demonstrated that NMs that possessed a negative zeta potential were more likely to enter plant leaves and translocate to unexposed tissues [29]. Therefore, negatively charged Se NMs used in this study have prospects for potential application in sustainable agricultural production.

### 3.2. The Growth of Tea Plant after the Foliar Application of Se NMs

After foliar spraying of 5, 10, and 50 mg·L^−1^ Se NMs, the tea growth was significantly enhanced. The results indicated that the net photosynthesis rate, transpiration rate, stomatal conductance, and intercellular CO_2_ concentration were elevated by Se NMs, and 10 mg·L^−1^ Se NMs exhibited the most effective outcomes, with increases of 48.7%, 31.2%, 28.1%, and 29.6%, respectively, regarding these photosynthetic parameters (Figure 2). Moreover, the concentrations of photosynthetic pigments, including chlorophyll a (Chla), chlorophyll b (Chlb), and carotenoids (Car) in the picked buds were significantly increased by 5 and 10 mg·L^−1^ Se NMs. As shown in Figure 2, 5 mg·L^−1^ Se NMs increased the content of Chla, Chlb, and Car by 58.8%, 39.3%, and 73.6%, respectively, and 10 mg·L^−1^ Se NMs improved Chla content by 14.7%. Additionally, the changes in the synthesis of the photosynthetic products after Se NMs spraying were measured. The results showed that the soluble sugar contents of picked young leaves treated with 10 mg·L^−1^ Se NMs were 30.3% higher than those of the unexposed control (Figure 2). The above results indicated that Se NMs enhanced the photosynthesis of tea plants, suppling more energy for plant growth and driving the photosynthates from the older leaves to the younger bud tips.

### 3.3. The Yield and Quality of Tea Plant after the Foliar Application of Se NMs

Tea buds were picked and weighed after Se NMs exposure. Clearly, the surface area of 100-bud FDDB corresponding to the treatments with 5 and 10 mg·L^−1^ of Se NMs were bigger than those in the unexposed control and the plants treated with 50 mg·L^−1^ of Se NMs (Figure 3A,B). Se NMs increased both fresh and dry 100-bud weights, and the highest 100-bud weight was detected at 10 mg·L^−1^ of Se NMs (24.3% and 28.8% increase, respectively) (Figure 3C). Particularly, the total fresh yield of FDDB increased by 21.9% upon 10 mg·L^−1^ Se NMs exposure (Figure 3D).

The quality characteristics, including theanine, catechins, and caffeine, of FDDB tea buds are important bioactive components. When compared with the unexposed control treatment, the theanine contents in the 5, 10, and 50 mg·L^−1^ Se NMs treatments were significantly higher (*p* < 0.05). Consistent with the growth parameters, the highest theanine content was observed in the 10 mg·L^−1^ Se NMs treatment, which was significantly 8.6–36.2% higher than that in the other treatments (Figure 3E). LC–MS/MS was used to determine the six common catechins and caffeine content. The EGCG contents in the picked tea buds of all the Se NMs (5, 10, 50 mg·L^−1^) treatments with were 51.2%, 53.9%, and 46.1% higher than that of the control, respectively (Figure 3E). The contents of ester or non-ester catechins in the Se NMs treatments were also significantly increased, and the 10 mg·L^−1^ Se NMs treatment expressed the highest catechins content (10.8 mg·g^−1^) (Figure 3G). The contents of caffeine from the 10 mg·L^−1^ and 50 mg·L^−1^ Se treatments were 30.8% or 32.4% higher, respectively, than that of the control (Figure 3E). 

Tea polyphenols and catechins determine the color and aroma of tea, which is directly related to the sensory quality and physiological health function of tea [7]. Li et al. found that Se NMs (10 mg·L^−1^) markedly enhanced the protein, soluble sugar, carotenoid, tea polyphenols, and catechins contents [18]. Foliar-applied NMs can reach the leaf surface, enter through stomatal openings or the base of the trichomes, and then travel to various tissues [30]. It has been shown that Se accumulates in plant leaves and is metabolized in chloroplasts [19,31,32]. In this study, Se NMs treatment significantly increased the contents of Se: the Se content in tea buds reached 0.09–0.31 mg·kg^−1^ among all Se treatments, and Se concentration in tea buds treated with 10 mg·L^−1^ Se NMs was 0.27 mg·kg^−1^ (Figure 3I). Therefore, we suggested that Se NMs can be uptaken by plants in the form of nanoparticles, thus resulting in a significant increase in the Se content of tea buds. Se is one of the essential trace elements of the human body, and the human body cannot directly synthesize and store Se for long periods of time. Dietary supplement of Se is considered to be a relatively safe and effective method of obtaining this trace element. The consumption of Se-rich foods can enhance human immunity and antioxidant capacity, delay aging, prevent diabetes, and reduce blood sugar [33,34]. In addition, nutrient elements including Na, P, K, Mg, Zn, and Mn were highly elevated by 30.7%, 18.5%, 15.4%, 17.1%, 39.3% and 33.7%, respectively, upon treatment with 10 mg·L^−1^ Se NMs (Figure 3H,I). Therefore, it is concluded that the leaf application of Se NMs is both beneficial and safe. As described above, among the applied doses, treatment with 10 mg·L^−1^ Se NMs showed the best effect in regards to improving the photosynthetic products and bud tip quality of FDDB.

### 3.4. Foliar Applied Se NMs Improve the Synthesis and Transport of Catechins and Theanine in Tea Buds

Based on the above experiments, the concentration of Se NMs (10 mg·L^−1^), the same ion concentration employed in common conventional commercial fertilizer (containing 46% Na_2_SeO_3_) used in agriculture, was screened to further explore the influence of different treatments on the growth and quality of tea buds. According to the experimental results (Appendix A), the contents of theanine, total catechins, ester or non-ester catechins, and caffeine in tea buds were significantly increased after Se NMs exposure compared to the levels in unexposed control group. However, there were no significant differences in the theanine and caffeine contents between the unexposed group and the ionized control group. Thus, 10 mg·L^−1^ Se NMs was selected as the optimal dose for the subsequent experiments.

Moreover, the metabolite profiles of the tea buds after the Se NMs treatments were different from those of the unexposed control. As shown in Figure 4A, samples from CK treatment were mainly concentrated on the left side, while the 10 mg·L^−1^ Se NMs-treated samples were mainly gathered on the other side. This indicated that the metabolite profiles of the buds from the Se NMs treatments were different from those of the CK treatment. There are 15 metabolites related to FDDB quality which showed significant differences after 10 mg·L^−1^ Se NMs treatment, including 6 amino acids, 2 alkaloids, 4 polyphenols, and 3 organic acids (Figure 4B). These metabolites were critical intermediates of the major metabolic pathways, including the citrate cycle (TCA cycle), pyruvate metabolism, amino acid metabolism, and glutamine synthase/glutamate synthase (GS/GOGAT) cycle (Figure 4C). The photosynthetic process and amino acid metabolism pathways of plants are the main means for utilizing C and N. For example, the GS-GOGAT pathway can regulate N assimilation, and its products act as signals or precursors to further regulate primary and secondary metabolism in plants [35]. Accordingly, under Se NMs treatment, the total C in tea buds increased by 8.9%, and the total C and total N in tea roots increased by 5.5% and 22.6%, respectively (Figure 4D). These enhanced pathways could promote the subsequent synthesis of amino acids (glutamic acid, arginine, tryptophan, theanine, etc.) and antioxidant active substances (caffeine, catechins, etc.). The biosynthesis of important ingredients in tea—catechins—follows the flavonoid pathway, which is produced from phenylalanine by a series of enzyme catalyzations encoded by CsPAL, CsC4H, CsCHI, CsDFR, CsANS, CsANR, CsLAR, and UGGT [36]. In comparison with control, the relative expression of CsPAL, CsC4H, CsCHI, CsDFR, CsANS, CsANR, CsLAR, and UGGT was significantly up-regulated by 45.1%, 596.7%, 508.5%, 136.1%, 228.30%, 943.8%, 514.7%, and 619.1%, respectively, in tea buds after Se NMs treatment (Figure 4E), suggesting that the Se NMs applied on the leaves of the plants could promote the biosynthesis of catechins. The upregulation of CsTs, CsGs, and CsGOGAT, encoding enzymes that catalyze theanine, glutamine, and glutamate biosynthesis, in tea buds under Se NMs exposure were 224.1%, 138.8%, and 693.7%, respectively, greater than that in the control (Figure 4F). 

Other than the leaves, theanine is mainly synthesized in tea roots and subsequently transported by members of the tea plant-specific amino acid permease (*CsAAP*) family as inflow carriers to the shoots and leaves for accumulation [37]. The relative expressions of *CsAPP1*, *CsAPP2*, *CsAPP5*, and *CsAPP8* in tea roots were significantly upregulated by 52.0%, 17.4%, 276.0%, and 40.8%, respectively, after Se NMs treatment (Figure 4G). In addition, the relative expressions of *CsAPP1*, *CsAPP2*, *CsAPP5*, and *CsAPP6* were significantly upregulated in tea leaf slices and significantly downregulated in tea buds (Figure 4H). It is reported that *CsAAP6* plays vital roles in the loading of theanine in the phloem by mediating the exchange between the xylem and phloem of mature leaf veins during shoot development [38]. Here, the application of Se NMs may inhibit the expression of *CsAPP6* in tea buds, thereby reducing the transformation in new buds and increasing the accumulation of theanine in the tea buds. Additionally, Se NMs treatments can promote theanine transportation in both leaves and roots.

### 3.5. Application of NMs Changes the Diversity and Structure of Soil Microbial Communities

Tea quality depends on tea varieties characterized by different metabolite concentrations. However, it also has a complex relationship with soil texture, soil nutrient composition, and the presence of rhizosphere microorganisms and soil microorganisms [39]. Photosynthate transferred to the roots can further exudate into the rhizosphere in the form of low molecular weight, and these low molecular organic compounds can recruit specific bacteria to influence the structure and function of soil microbial communities, playing a key role in plant–microbe interactions [38,40]. The relative expressions of *CsSUT1*, *CsSUT2*, *CsSUT3*, and *CsSWEET1a* exposed to Se NMs were increased by 125.8%, 126.3%, 560.5%, and 374.3%, respectively (Appendix A). Sucrose transporter protein (SUT) and hexose and sucrose transporter protein (SWEET) are important for efficient silique loading and the distribution of photosynthetic products for root development [41,42], indicating that more sucrose can be transferred to the roots to supply tea tree growth and development. Significant levels of microbial population differences were observed between the Se NMs exposure treatments and the unexposed control. No significant difference was observed between groups in the Simpson and Shannon diversity index, which might be due to the low effect of 10 mg·L^−1^ Se NMs on the microbial diversity of tea plantation soils (Figure 5A). In addition, β-diversity was analyzed to reveal the differences in bacterial community composition. The first two principal components of PCoA accounted for 37.6% and 30.8% of the total data variance, respectively, and indicated that there were significant differences in bacterial community structure between Se NMs and unexposed treatments (Figure 5B).

The phylogenetic analysis of inter-root microorganisms after the application of Se NMs showed that Proteobacteria, Acidobacteria, and Actinobacteria were the dominant phylum in both the control and Se NMs-treated rhizosphere soils; included among these, Se NMs treatment significantly upregulated the abundance of Proteobacteria, Actinobacteria, WPS-2, and Verrucomicrobia and decreased the content of Actinobacteria, Bacteroidetds, and Firmicutes at the phylum level (Figure 5C). Previous studies have shown that foliar application of NMs also have the potential to regulate rhizosphere microorganisms [43,44]. Notably, Se NMs significantly enriched the relative abundance of *Acidibacter*, *Subgroup-13*, *Conexibacter*, and *Pedosphaeraceae* at the genus level by 50.9%, 71.4%, 54.1%, and 31.8%, respectively (Figure 5D). Among them, *Acidibacter*, *Roseiarcus*, *Acidothermus*, *Conexibacter*, and *Pedosphaeraceae* play positive roles in promoting soil phosphate solubilization, converting complex ammoniates in the roots into ammonium nitrogen (NH_4_^+^-N) for easy uptake and utilization, and combined nitrogen fixation, respectively [45,46,47,48,49,50,51]. Nitrogen is one of the essential elements for higher plants, playing a vital role in plant growth, development, photosynthesis, and metabolic regulation. Tea plants are high nitrogen consumers, and nitrogen directly or indirectly affects the metabolism, growth, and development of tea plants, which in turn greatly affects the quality of tea leaves. Studies have shown that inorganic nitrogen is the main source of nitrogen absorbed and utilized by plants, and that the roots of tea plants show a preference for ammonium salts (NH_4_^+^-N) compared to nitrate (NO_3_^−^-N) [52,53,54,55,56]. The content of NH_4_^+^-N and NO_3_^−^-N in tea roots significantly increased by 51.3% and 258.5%, respectively, after the application of Se NMs. However, the NH_4_^+^-N and NO_3_^−^-N content in the rhizosphere soil showed the opposite trends, and they were significantly decreased by 15.5% and 15.8%, respectively. This might be due to the fact that more nitrogen was taken up by the tea roots (Appendix A). The external NH_4_^+^-N entered the tea tree by the action of transporter proteins and was assimilated by the GS/GOGAT cycle, then transported to various parts of the plant to form different amino acids, such as theanine, which again explained the increase in theanine content due to the application of Se NMs [57]. Finally, Se NMs promoted the increase in quality and production of tea, and the long-term effects of Se NMs on tea plants should be studied in future research. On the other hand, we also need to further consider the potential accumulation of Se NMs in the soil and its effects on soil microbiota, groundwater contamination, and overall ecosystem health.

## 4. Conclusions

In the present work, field studies of tea plants showed that low concentrations (10 mg·L^−1^) of Se NMs could improve the nutritional level of green tea, as evidenced by the increase in tea-bud fresh weight, 100-bud weight, soluble sugar, theanine, tea polyphenols, and caffeine contents. In addition, we explored the mechanism of Se NMs to improve the quality of green tea. The results indicated that glucose metabolism and the expression of theanine and tea polyphenol synthesis genes were upregulated. NMs could improve the efficiency of photosynthesis and upregulate the hexose and sucrose transporter genes *CsSUT* and *CsSWEET*, facilitating the transport of photosynthetic products from the leaves downwards and improving the beneficial microbiota (*Roseiarcus*, *Acidothermus*, *Acidibacter*, *Conexibacter*, and *Pedosphaeraceae*). NMs promote the uptake and utilization of nitrogen and other nutrients in tea plants. In addition, the enhancement of nutrient levels by Se NMs was two orders of magnitude greater than that of the equivalent ion treatment. Overall, this study reveals the potential of Se NMs to improve the quality of green tea in low doses, while reducing environmental impact of high fertilizer inputs due to traditional agricultural practices, with great potential for the development of sustainable nano-agriculture.

## Figures and Tables

**Figure 1 nanomaterials-14-00681-f001:**
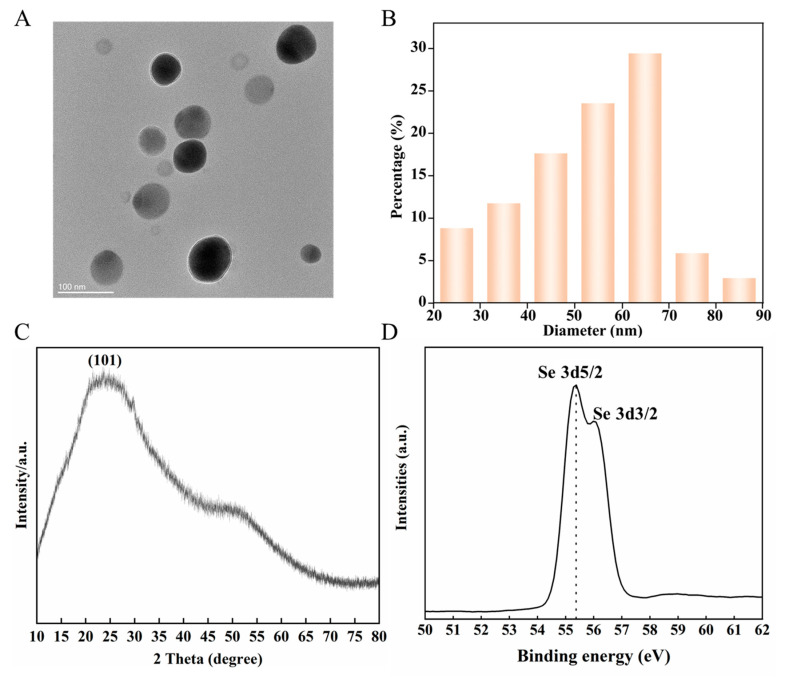
Characterization of synthesized and Se NMs. (**A**) TEM image; (**B**) size distribution of Se NMs; (**C**) XRD pattern; (**D**) XPS diagram of Se NMs.

**Figure 2 nanomaterials-14-00681-f002:**
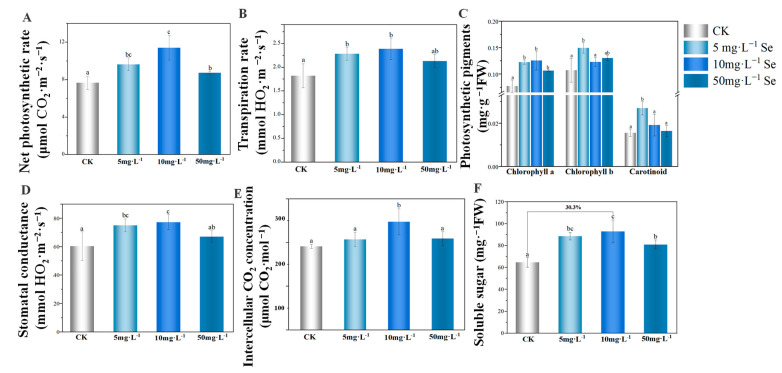
(**A**) Net photosynthetic rate; (**B**) transpiration rate; (**C**) photosynthetic pigments; (**D**) stomatal conductance; (**E**) intercellular CO_2_ concentration; and (**F**) soluble sugar in FDDB tea buds treated with 5, 10, and 50 mg·L^−1^ Se NMs. Different letters represent significant differences (Duncan test; *p* < 0.05).

**Figure 3 nanomaterials-14-00681-f003:**
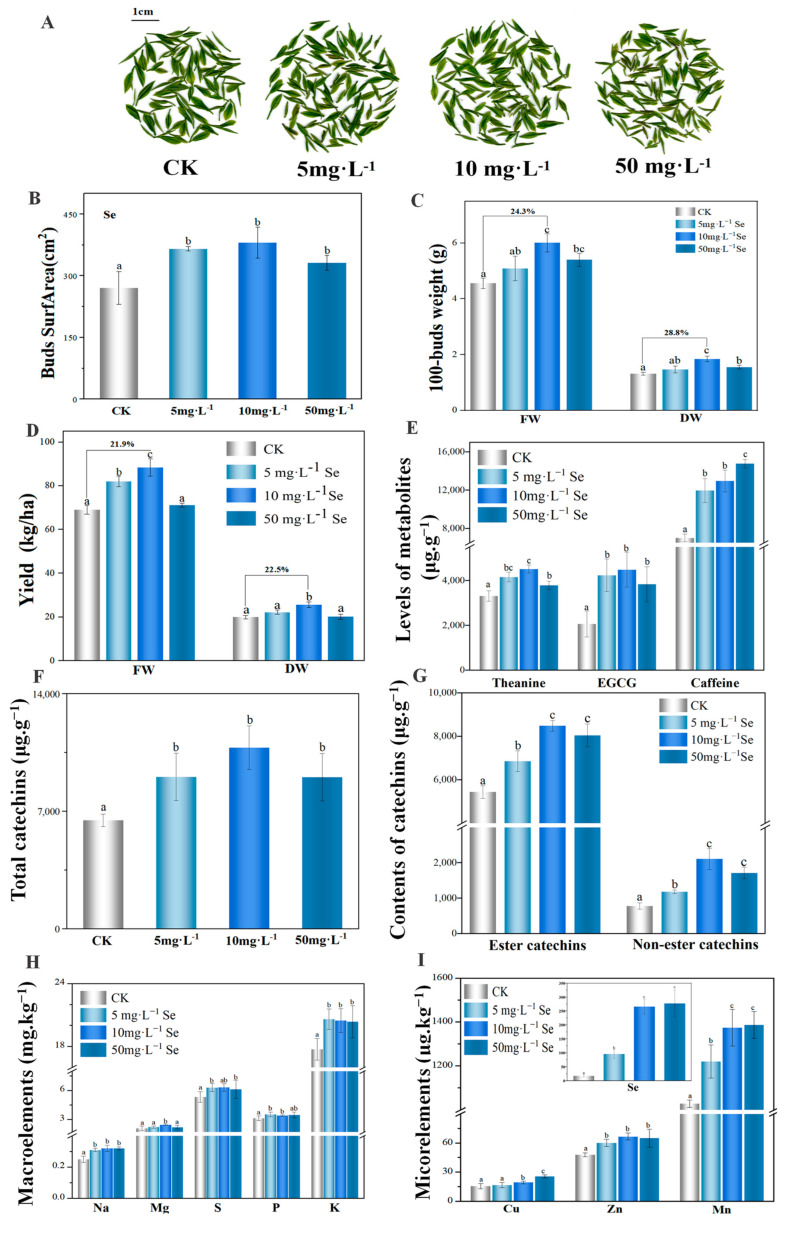
(**A**) Image of picked young tea bud tips and their production and quality under Se NMs treatments: (**B**) buds surface area; (**C**) 100-bud fresh weight (FW) and dry weight (DW); (**D**) total yield; (**E**) contents of theanine, EGCG, and caffeine; (**F**) total catechins content; (**G**) ester and non-ester catechins content; (**H**) macroelement levels; and (**I**) microelement levels. Different letters indicate significant differences according to Duncan’s test, with *p* < 0.05.

**Figure 4 nanomaterials-14-00681-f004:**
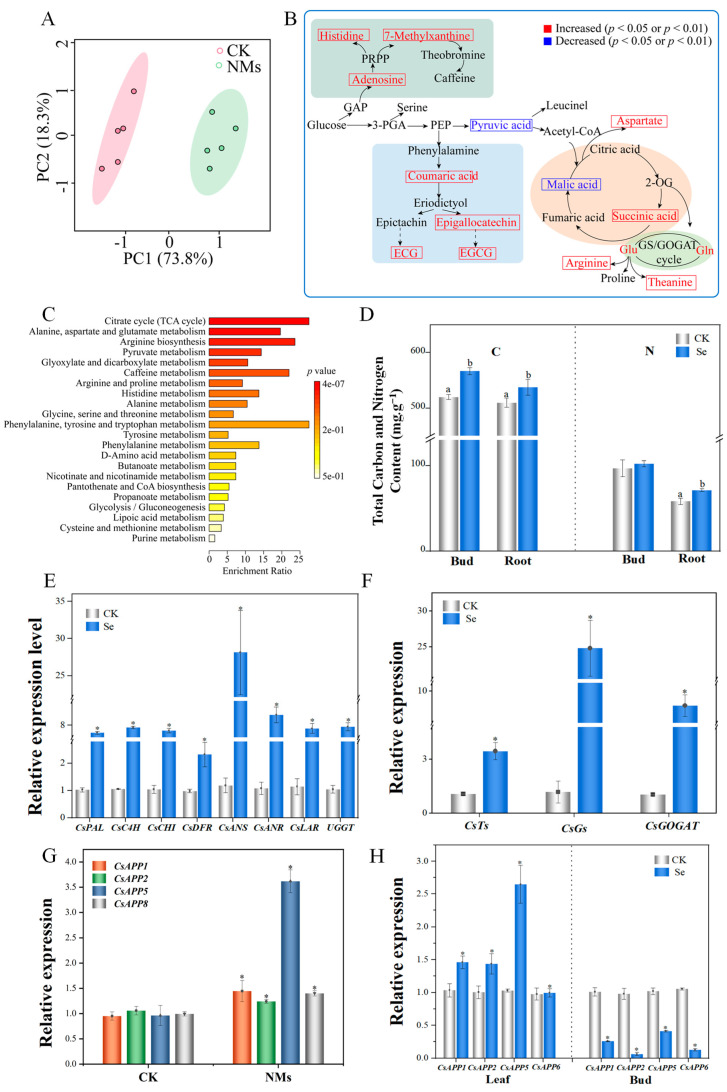
(**A**) Principal components analysis (PCA) of metabolites in tea buds after application of Se NMs; (**B**) changes in the metabolites profiles and metabolic pathways in the tea buds of FDDB under Se NMs treatment; (**C**) enrichment analysis of tea bud metabolic pathways under Se NMs; (**D**) total carbon (C) and nitrogen (N) content in tea buds and roots; (**E**) relative expression of catechins in tea leaves and (**F**) theanine synthesis genes in tea leaves; (**G**) relative expression of theanine transport in tea roots and (**H**) in tea leaves and buds. Different letters represent significant differences, and asterisks indicate * *p* < 0.05.

**Figure 5 nanomaterials-14-00681-f005:**
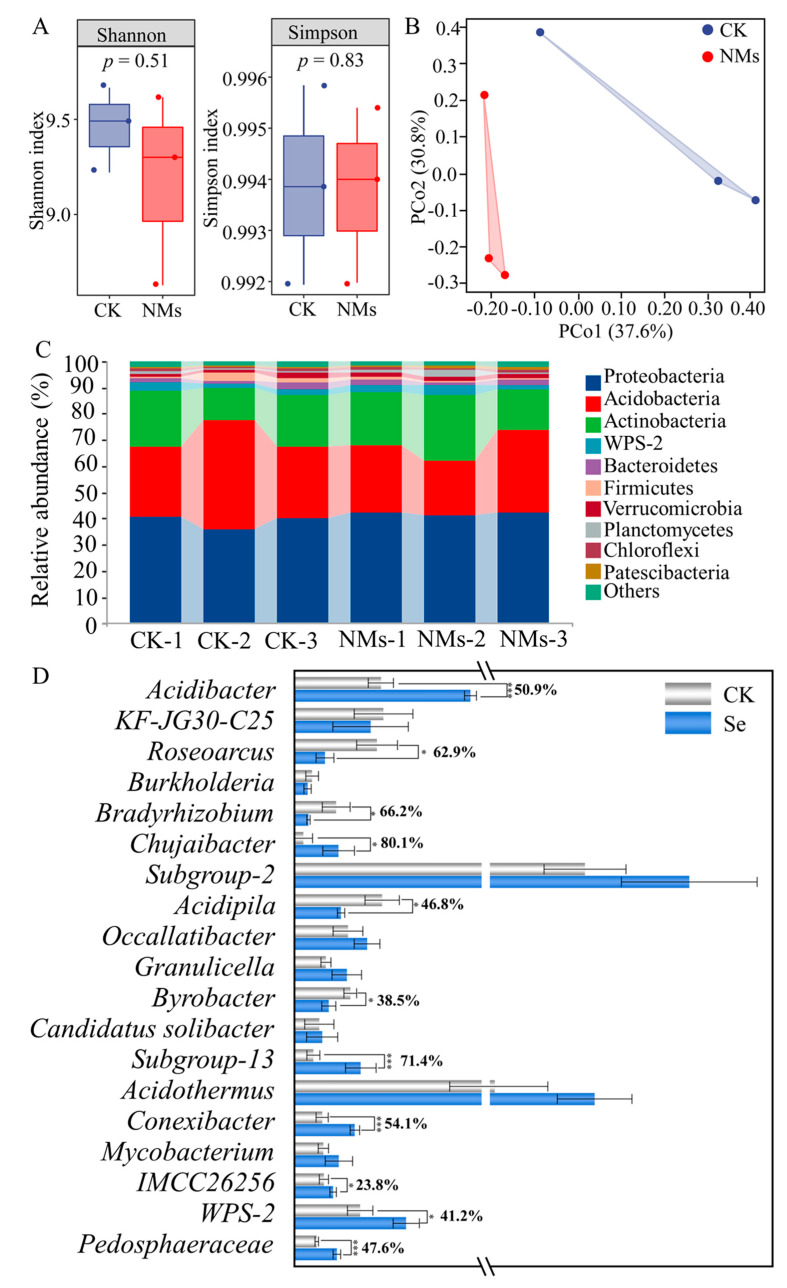
The altered rhizosphere microbiome response to the application of Se NMs. (**A**) Alpha diversity measured by the Simpson and Shannon index; (**B**) principal coordinates analysis (PCoA); (**C**) relative abundance of dominant rhizosphere bacterial communities at the phylum level; and (**D**) analysis of relative abundance at the genus level under Se NMs treatment. Asterisks indicate * *p* < 0.05 and *** *p* < 0.001, respectively.

## Data Availability

The data presented in this study are available on request from the corresponding author.

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
