# Peer review of "Selenium Nanomaterials Enhance the Nutrients and Functional Components of Fuding Dabai Tea"

_nanomaterials, 2024, doi:10.3390/nano14080681_

Round 1

Reviewer 1 Report

Comments and Suggestions for Authors

This work examines the effect of the introduction of selenium nanomaterials on the characteristics of tea plants: biomass, as well as the content of nutritional components such as soluble sugar, theanine, polyphenols and caffeine, and some elements (Mg, P, S, K, Mn). The use of nanomaterials in agricultural technologies has been actively researched recently, and therefore this work may be of scientific interest. However, a number of errors and inaccuracies were identified in the text of the manuscript that don’t allow the work to be accepted in this form. Below is a list of comments and suggestions that will improve the quality of this work.

1. The first sentence of the abstract needs to be rewritten, because it doesn’t reflect the essence of this work.

2. Information on the characteristics of the selenium nanomaterials used are extremely important and should be included in the main text of the manuscript, and not in the supplementary.

3. The authors should also expand the introduction, supplement it with modern works on the bioactive properties of selenium nanoparticles: antioxidant, antibacterial, cytoprotective, etc. (10.3390/ma16155363; 10.3390/ma16145164).

4. In line 35, the data provided must be accompanied by a full link to the source, as well as in line 39.

5. Please correct in line 91 to × g

6. Line 375 –  taxon name in Latin cannot be indicated with a small letter.

7. Visualization in some cases is of poor quality, the axes of some graphs are difficult to read (Figure 3. B, C)

8. There is no explanation in the text for “CK” (lines 270, 273)

Reviewer 2 Report

Comments and Suggestions for Authors

This study on the effects of selenium nanomaterials (Se NMs) on the synthesis of fat-reducing components in Fuding Dabai (FDDB) tea seems comprehensive and promising, but it also faces several challenges. Authors will need to add the answers to the following questions to the manuscript:

  1. What are the long-term effects of Se NMs on tea plants and the quality of tea produced? Did the authors monitor changes in soil composition, plant health, and the sustainability of Se NM application over multiple growing seasons?
  2. Did the authors evaluate any potential accumulation of selenium in the soil and its effects on soil microbiota, groundwater contamination, and overall ecosystem health?
  3. Authors should include toxicological assessments to determine if the increased selenium levels in tea pose any health risks to consumers.
  4. How could the authors determine the optimal dosage and application methods of Se NMs for maximizing the desired effects on tea quality while minimizing any negative impacts on plant health and the environment?
  5. Authors need to assess the economic feasibility of Se NM application in tea cultivation by evaluating the cost-effectiveness of Se NM production, application, and potential market demand for selenium-enriched tea products.
  6. Make the Figure 1c and Figure 2 legends larger in size and readable.
  7. Figures 3b and 3c are not readable.
  8. Figures 4d and 4E need to be readable.
  9. How did the authors ensure that the results of their study are replicable across different tea cultivars, growing conditions, and geographical regions?
  10. Authors will need to explore how Se NM application can be integrated with existing agricultural practices while maintaining tea quality and authenticity.

Comments on the Quality of English Language

As above.

Reviewer 3 Report

Comments and Suggestions for Authors

I had the pleasure of reviewing your manuscript "Selenium Nanomaterials Enhanced the Nutrients and Functional Components of Fuding Dabai Tea". The authors present a study on the effects of selenium nanomaterials (Se NMs) on the synthesis of fat reducing components in Fuding Dabai (FDDB) tea, which is an interesting topic. However, the manuscript should be further improved. In order to improve the manuscript, the following comments are made.

1.     Abstract section looks incomplete and not properly arrange. An abstract summarizes the major aspects of the entire paper in a prescribed sequence that includes: 1) the overall purpose of the study and the research problem(s) you investigated; 2) the basic design of the study; 3) major findings or trends found as a result of your analysis; and, 4) a brief summary of your interpretations and conclusions.

2.     Introduction: There are more than 900 articles on the biological activity of selenium nanoparticles and their effect on obesity. What new information does your article add to the literature? Authors should continue to emphasise the novelty of their work.

3.     What was the pH value at which the hydrodynamic diameter and zeta potential of selenium nanoparticles solutions were measured? Was a background electrolyte used during these measurements?

4.     I think the authors should include a TEM image of selenium particles in the body of the article, as well as a graph showing the particle size distribution. How many particles were counted to determine the average particle size of selenium? It should be mentioned in the Materials and Methods.

5.     All the figures should be improved, including the definition and aesthetic.

6.     The XPS-spectra of Se NMs provided in supplementary materials must be analyzed considering the spin-orbit components Se 3d3/2 and Se 3d5/2.

7.     In this paper there was a study of soil microbiocenoses. However, as far as I know, tea bushes are treated using the foliar method. Is it appropriate to study the microbiocenosis with this type of treatment and does it change due to other factors?

8.     The results presented in the microbiocenosis study  suggest an improvement in nitrogen metabolism. However, there is no information on the exchange of phosphorus and potassium in the rhizosphere, which are essential macronutrients for plant growth and development.

9.     Please check for significant figures in the numbers throughout the study. Match them with the confidence interval.

10. An improvement of the discussion is needed also in relation to similar or analogous results from literature and the explanation of possible differences.

Round 2

Reviewer 1 Report

Comments and Suggestions for Authors

The authors of the manuscript have improved the work significantly. All necessary improvements have been made.

Reviewer 2 Report

Comments and Suggestions for Authors

N/A

Comments on the Quality of English Language

As above.